# Ceftolozane-Tazobactam Treatment of Hypervirulent Multidrug Resistant *Pseudomonas aeruginosa* Infections in Neutropenic Patients

**DOI:** 10.3390/microorganisms8122055

**Published:** 2020-12-21

**Authors:** Paolo E. Coppola, Paolo Gaibani, Chiara Sartor, Simone Ambretti, Russell E. Lewis, Claudia Sassi, Marco Pignatti, Stefania Paolini, Antonio Curti, Fausto Castagnetti, Margherita Ursi, Michele Cavo, Marta Stanzani

**Affiliations:** 1Institute of Hematology “Seràgnoli”, IRCCS-Azienda Ospedaliero Policlinico Sant’Orsola-Universitaria di Bologna, 40138 Bologna, Italy; paoloelia.coppola@studio.unibo.it (P.E.C.); chiara.sartor2@unibo.it (C.S.); stefania.paolini@aosp.bo.it (S.P.); antonio.curti2@unibo.it (A.C.); fausto.castagnetti@gmail.com (F.C.); margherita.ursi@studio.unibo.it (M.U.); michele.cavo@unibo.it (M.C.); 2Microbiology, IRCCS-Azienda Ospedaliero Policlinico Sant’Orsola-Universitaria di Bologna, 40138 Bologna, Italy; paolo.gaibani@unibo.it (P.G.); simone.ambretti@aosp.bo.it (S.A.); 3Infectious Diseases, IRCCS-Azienda Ospedaliero Policlinico Sant’Orsola-Universitaria di Bologna, 40138 Bologna, Italy; russeledward.lewis@unibo.it; 4Department of Medical and Surgical Sciences (DIMEC)- Università di Bologna, Alma Mater Studiorum, 40138 Bologna, Italy; 5Radiology, IRCCS-Azienda Ospedaliero Policlinico Sant’Orsola-Universitaria di Bologna, 40138 Bologna, Italy; claudia.sassi3@unibo.it; 6Department of Diagnostic and Experimental Medicine Specialty (DIMES)- Università di Bologna, Alma Mater Studiorum, 40138 Bologna, Italy; marco.pignatti@unibo.it; 7Plastic Surgery, IRCCS-Azienda Ospedaliero Policlinico Sant’Orsola-Universitaria di Bologna, 40138 Bologna, Italy

**Keywords:** *Pseudomonas aeruginosa*, ceftolozane/tazobactam, hypervirulent, skin and soft tissue, neutropenia, hematological malignancy

## Abstract

The effectiveness of ceftolozane/tazobactam for the treatment of infections in neutropenic patients caused by hypervirulent multidrug-resistant (MDR) *Pseudomonas aeruginosa* has not been previously reported. We identified seven cases of MDR *P. aeruginosa* infection in neutropenic patients over a four-month period within the same hematology ward. Four cases were associated with rapid progression despite piperacillin-tazobactam or meropenem therapy, and three patients developed sepsis or extensive skin/soft tissue necrosis. In three of the four cases, patients were empirically switched from meropenem to ceftolozane/avibactam before carbapenem susceptibility test results were available, and all four patients underwent extensive surgical debridement or amputation of affected tissues and survived. Further investigation revealed a common bathroom source of MDR *P. aeruginosa* clonal subtypes ST175 and ST235 that harbored genes for type III secretion system expression and elaboration of ExoU or ExoS exotoxin. We conclude that ceftolozane/tazobactam plus early source control was critical for control of rapidly progressing skin and soft infection in these neutropenic patients caused by highly virulent ST175 and ST235 clones of MDR *P. aeruginosa*.

## 1. Introduction

In neutropenic patients, breakthrough infections with *Pseudomonas aeruginosa* can present as rapidly progressing pneumonia, bacteremia, and sepsis associated with mortality rates of 60–80% [1]. The involvement of other internal organs, limbs, and genitalia is less common but often requires timely surgical intervention in additional to broad-spectrum antibiotics [2,3]. Although the infection often arises endogenously following mucotoxic chemotherapy during neutropenia, exogenous acquisition and horizontal transmission of multidrug-resistant (MDR) strains have been documented via contaminated sinks, toilets, and bathrooms [4,5,6].

Carbapenems are frequently used as empiric therapy for febrile neutropenia in centers with a high prevalence of extended-spectrum beta-lactamase Enterobacterales, creating pressure for the selection and dissemination of carbapenem-resistant strains [7]. Comparative genomic analysis of MDR *P. aeruginosa* isolates from patients with bloodstream infection has identified dominant clonal sub-types of carbapenem-resistant *P. aeruginosa* (e.g., ST175, ST235) that express virulence attributes such as the type III secretion system (TTSS) [8,9,10]. This secretion system injects exotoxins, including ExoS, ExoT, ExoU, or ExoY phospholipases, into eukaryotic cells, resulting in distinct patterns of host tissue injury depending on the high-risk clone and site of infection, with ExoU exotoxin associated with the greatest impact on bacterial virulence in human hosts [11]. Therefore, the selection of high-risk MDR and carbapenem-resistant *P. aeruginosa* strains may favor some-high risk clonal lineage with enhanced virulence [10].

Ceftolozane/tazobactam is a novel cephalosporin with activity against gram-negative bacteria, including *P. aeruginosa* and *Escherichia coli* [12,13,14], and has a safety profile similar to other cephalosporins. Ceftolozane/tazobactam is less susceptible to bacterial cell efflux and degradation by multiple classes of beta-lactamases, with reported susceptibility rates for *P. aeruginosa* of 90% (including carbapenem-resistant strains) [12,14,15]. Therefore, ceftolozane/tazobactam is an appealing monotherapy treatment option compared to older antibiotics that are often used in combination with more toxic aminoglycosides or polymyxins. Although more clinical data in the hematology population is needed, ceftolozane/tazobactam appears to be highly active in the treatment of complicated intra-abdominal and urinary tract infections [16,17], pneumonia [18], and difficult-to-treat (DTR) *P. aeruginosa* [19]. In a recent small single-center case-control study, ceftolozane/tazobactam was well tolerated and at least as effective as other alternatives for *P. aeruginosa* infection in patients with hematologic malignancies, including neutropenic patients [20]. However, it is unknown whether early ceftolozane/tazobactam treatment can improve outcomes of neutropenic patients infected with highly virulent MDR clones of *P. aeruginosa*.

Herein, we report a monocentric outbreak of four cases of hypervirulent MDR *P. aeruginosa* infection associated with a common identified bathroom source that occurred in the same hematology unit in Italy over a four-month period. All patients, except one, shared the same bathroom, and three patients were treated with ceftolozane/tazobactam for rapidly progressing clinical infection.

## 2. Case Descriptions

Collection of data related to this report was approved by the institutional ethics committee as part of routine infection control surveillance in accordance with the ethical standards laid down in the 1964 Declaration of Helsinki and its later amendments and Italian law. All patients described in this report provided informed consent to present their case history and medical images.

### 2.1. Case #1—Index Case

A 43-year-old man with acute lymphoblastic leukemia (ALL) in complete remission (CR) was admitted in early February to room C (Figure 1) to receive a fourth cycle of consolidation chemotherapy. Ten days after the end of chemotherapy, while neutropenic, he developed fever with severe sepsis, and increased procalcitonin (PCT, 40 ng/mL) and C-reactive protein (CRP, 12 mg/dL). Blood cultures were positive for *P. aeruginosa* that was susceptible to piperacillin/tazobactam, amikacin, cefepime and meropenem. The patient was isolated in a single room with a personal bathroom.

Unfortunately, the patient remained persistently febrile despite receiving 18 g of piperacillin/tazobactam by continuous infusion for >48 h. After switching the patient to 2 g of meropenem every 8 h administered in 6-h infusions plus amikacin 15 mg/kg/day, he improved rapidly and was discharged home without complications.

### 2.2. Case #2

A 45-year-old man affected by newly diagnosed high-risk acute myeloid leukemia (AML) was admitted January 28 to room D (Figure 1) for induction chemotherapy with fludarabine, cytarabine, and idarubicin. Seven days after the end of the chemotherapy, he presented fever with severe sepsis and began empiric antibiotic therapy with piperacillin/tazobactam. At the fever onset, he reported pain localized under his right toe where a lenticular lesion was visible. We suspected the patient had been washing his feet in the bathroom bidet. After 48 h, blood cultures (BCs) were positive for *P. aeruginosa* while the skin lesion had now rapidly evolved into blisters (Figure 2A) and the patient developed excruciating pain at palpation of the toe, highly suspicious for necrotizing fasciitis.

Antibiotic susceptibility testing revealed that an isolate was sensitive only to amikacin, colistin, and ceftolozane/tazobactam. His antibiotics were changed to 1.5 g of ceftolozane/tazobactam every 8 h, 9 million international units (MIU) colistin daily in two divided doses daily following a 9 MIU loading dose, and 15 mg/kg/day of amikacin.

A computed tomography (CT) scan of the right leg was performed and showed no signs of necrotizing fasciitis (Figure 2C). The right foot blisters were drained, but no purulent colliquated material was recovered (Figure 2B). Unfortunately, the fasciitis worsened over the ensuing 48 h on antibiotic therapy, and a follow-up CT scan taken 5 days from previous imaging showed infection progressing to the subcutaneous tissue and muscles up to the right lower limb with associated lymphangitis. An amputation of the right leg was deemed necessary. A few days after the operation, the patient improved with resolution of pain and the lymphangitis with progressively reduced erythema. The patient received 16 days of additional antibiotic therapy and recovered from neutropenia 7 days after infection onset, with the aid of granulocyte colony stimulating factor. The patient was discharged home in complete remission of his underlying malignancy without antibiotic therapy.

### 2.3. Case #3

A 60-year-old man, affected by newly diagnosed AML, was admitted on February 19 to room C (Figure 1). The patient received induction chemotherapy, to which he was refractory, followed by a cycle of rescue chemotherapy. After 12 days from the end of the second cycle, he developed fever with severe sepsis complicated by initial balanoposthitis. Empiric antibiotic therapy was started with meropenem, but the lesion evolved rapidly with painful erythema and edema of pubis, penis, scrotum, perineal, and inguinal lymphadenopathy.

Superficial necrosis similar to Fournier’s gangrene (Figure 3A–C) was present on the penis. An ultrasound revealed soft tissue imbibition, signs of epididymitis, and conserved parenchyma of penis and testicles. A chest and abdominal CT scan were also performed that did not identify other infection localization. After 2 days from the fever onset, the BCs turned positive for an MDR *P. aeruginosa* sensitive to meropenem, imipenem, and ceftolozane/tazobactam.

Considering the rapid progression of the infection and previous cases, we immediately started antibiotic therapy with ceftolozane/tazobactam, colistin, amikacin, meropenem, and clindamycin, with the latter two antibiotics later discontinued within 48 h. The patient improved rapidly with reduced fever and stabilization of erythema and necrosis (Figure 3D). To accelerate the healing of necrotic tissue, the urologist initiated hyperbaric therapy sessions. After eight 1-h hyperbaric therapy sessions and continued antibiotic therapy (ceftolozane/tazobactam and meropenem), edema, lymphadenopathy, and erythema resolved. A dry necrotic dorsal eschar persisted (Figure 3E) that was treated by plastic surgeons with debridement (Figure 3F), full thickness skin graft (Figure 3G), and negative pressure dressing (Figure 3H) for 5 days to improve skin graft take (Figure 3I). The patient achieved a CR for his hematological malignancy and was discharged after 16 days from the beginning of the infectious episode.

### 2.4. Case #4

A 64-year-old woman affected by B-follicular lymphoma was admitted on April 4 in room G (Figure 1) to undergo autologous stem-cell transplantation. The transplant was performed on April 24, and the following phase of neutropenia was complicated 10 days later by an episode of septic shock with hepatic failure, serum hyperosmolarity, hypotension, and atrial fibrillation. *P. aeruginosa* was detected within 24 h of blood culture. Empiric antibiotic therapy was promptly changed from meropenem plus amikacin to ceftolozane-tazobactam. The patient developed multiorgan failure (MOF) and was transferred to the ICU where she was clinically improved.

After 48 h, she was transferred back to the hematology unit to continue ceftolozane/ tazobactam and clinical support. The patient continued ceftolozane/tazobactam for 14 days and then was discharged home after 19 days in complete hematologic remission.

### 2.5. Cases #5–7

Three more patients hospitalized in the rooms C and D developed less severe infections by *P. aeruginosa* during the same time. Case 5 was a 63-year-old male affected by multiple myeloma who presented an asymptomatic urinary infection by *P. aeruginosa* sensitive to meropenem and amikacin at the beginning of March, which did not require treatment. Case 6 was a 67-year-old man with mantle B-cell lymphoma who presented with *P. aeruginosa* sensitive only to colistin and ceftolozane-tazobactam that resolved quickly with ceftolozane-tazobactam and neutrophil recovery. Patient 7, a 72-year-old man affected by AML, who was hospitalized in room D during February and March, died at an outside hospital in May for a severe sepsis caused by *P. aeruginosa*. He did not receive treatment with ceftolozane/tazobactam.

Because nearly all the *P. aeruginosa* infections involved patients admitted to rooms C and D (Table 1), we performed environmental microbiological surveys in the shared bathroom between the two involved rooms, checking for 20 sites of possible common source in the shared bathrooms, rooms, and common areas. *P. aeruginosa* was isolated in the water trap filter and in the tap water of the bathroom bidet. After the filter removal and cleaning, no other *P. aeruginosa* infections were detected in this unit after 8 months of follow-up.

## 3. Microbiology

*P. aeruginosa* strains were identified by MALDI-TOF MS assay (Bruker Daltonics, Leipzig, Germany). Antimicrobial susceptibility testing was performed by Microscan (Beckman Coulter, Brea USA) and confirmed by MIC test strip (Liofilchem, Roseto degli Abruzzi, Italy). MIC results were interpreted following EUCAST clinical breakpoints v9.0. Previous cumulative susceptibility reports from our institution indicated that 92% of all tested *P.* aeruginosa isolates (96% from blood samples) were susceptible to ceftolozane/tazobactam. Whole-genome sequencing was performed as previously described [9]. Briefly, libraries were prepared by the Nextera XT sample preparation kit and sequenced using the Illumina MiSeq platform (Illumina, San Diego, USA) with a 2 × 250 paired-end run. All read sets were evaluated by FastQC software and then assembled with SPAdes v.3.10 with careful settings.

Unfortunately, genomes from clinical strains isolated from patients 1 and 2 were not included in the analysis because isolates were not available at the time of sequencing. Assembled genomes were screened for known antimicrobial resistance, sequence type (ST), and phage regions by CGE server (https://cge.cbs.dtu.dk/services/MLST/), Pubmlst(https://pubmlst.org/bigsdb?db=pubmlst_paeruginosa_seqdef) and PHAGE (http://phast.wishartlab.com) web tools. A core genome single nucleotide polymorphism (SNP) phylogeny was generated by Parsnp using the complete genome of strain PAO1 as reference. The phylogenetic tree was edited and visualized using FigTree software v1.4.3 (http://tree.bio.ed.ac.uk/software/figtree/). The whole-genome project is available at EMBL/ENA/EBI under study number PRJEB34087.

A summary of phenotypic and genotypic traits of *P. aeruginosa* strains analyzed in this study are shown in Table 2 and Table 3. Antimicrobial susceptibility profiles of clinical *P. aeruginosa* strains demonstrated patient isolates from cases 3 and 4 were susceptible to ceftolozane/tazobactam, ceftazidime/avibactam, and colistin, but resistant to gentamicin, ceftazidime, and piperacillin/tazobactam. In addition, 4/5 *P. aeruginosa* clinical isolates were resistant to imipenem and cefepime.

At the same, time, environmental *P. aeruginosa* strains isolated from the bidets were resistant against all antimicrobials except colistin. Genetic analysis showed that PABO48 and PABO68 (strains isolated from cases 4 and 6) harbored the same genes coding for antimicrobial resistance to β-lactams, aminoglycoside, fluoroquinolone, and sulfonamide and possessed similar prophage regions. Similarly, PABO62 (strain isolated from patient 3) and PABO93 and PABO94 (strains isolated from the water and filter in bathroom C-D) shared a similar genetic antimicrobial gene pattern. Of note, the two environmental *P. aeruginosa* strains harbored carbapenemase gene blaVIM-1.

MLST analysis demonstrated that PABO48 and PABO68 strains belonged to ST175, while PABO62 and two environmental strains (i.e., PABO93 and PABO94) belonged to the ST235. A phylogenetic tree constructed based on the core genome SNPs analysis of *P. aeruginosa* genomes revealed that clinical isolates PABO68 and PABO48 clustered within the ST175 group, whereas clinical strain PABO62 and environmental isolates PABO93 and PABO94 grouped closely within the ST235 group (Figure 4). A deeper examination of the genomic relationship between *P. aeruginosa* strains included in this study demonstrated that PABO48 and PABO68 isolates clustered closely within the ST175 group and PABO62, PABO93, and PABO94 strains formed a monophyletic group within ST235 (Appendix A).

Finally, we identified genes in both patient and environmental isolates reported to influence the virulence of *P. aeruginosa* infections. These included genes involved in (i) alginate production (*alg* and *muc*); (ii) flagella (*fle, flg*, *flg)*, lipopolysaccharide *(waa)*, and type IV pili expression (*chp, fim, pil*, and *xcpA/pilD*); (iii) rhamnolipid production (*rhlA* and *rhlB*), iron uptake through pyochelin (*fpt, pch),* and pyoverdine (*fpvA, prd*) siderophores; (iv) pyocyanin pigment production (*phnA/B*), (v) aprA, lasA, lasB protease secretion; (vi) quorum molecule responsive transcriptional regulators (Las1, LasR, Rhll, rh1R), (vii) Hcp secretion island-1 encoded type VI secretion system (*iclp, ppk, icm, dot, his, hcp, lipd, vrg, fha, tag);* (viii) type-III secretion systems (*psc, pcr, exs, pop*); and (ix) Xcp extracellular protein secretion system (*exp* gene family). All sequences isolates expressed the capacity for producing plcH and ToxA exotoxin. However, ST175 isolates BO48 and BO68 also harbored exoT, exoU and exoY and exoS, exoT, and exoY, respectively. Among strains belonging to clonal subtype ST235, the patient isolate BO62 also harbored exoS, ExoT, and exoY, while the isolates recovered from the bidet water (BO93) and filter (BO94) also contained exoT, exoU, and exoY and exoS, exoT, exoU, and exoY, respectively.

## 4. Discussion

This limited experience suggests that ceftolozane/tazobactam may be an effective treatment option for hypervirulent MDR *P. aeruginosa* infections in neutropenic patients that were likely acquired from a common bathroom source. Aggressive source control requiring amputation (Case 2) and hyperbaric therapy to enhance tissue healing and accelerated regeneration (Case 3) were used in combination with ceftolozane/tazobactam-based regimens.

Clinical data concerning the efficacy of ceftolozane/tazobactam for the treatment of severe *P. aeruginosa* infections in patients with hematological malignancies is limited. In a small case-control study, Fernández-Cruz et al. compared the outcomes of 19 neutropenic patients with *P. aeruginosa* who received ceftolozane/tazobactam monotherapy versus 38 controls who received other antibiotics [20]. The authors found no significant differences in clinical cure at day 14 (89.5% versus 71.1%; *p* = 0.183) or recurrence (15.8% versus 10.5%; *p* = 0.675). However, 30-day mortality was lower among ceftolozane/tazobactam-treated patients (5.3% versus 28.9%; *p* = 0.045). The authors concluded that ceftolozane-tazobactam was well tolerated and as effective as other alternatives for *P. aeruginosa* infection in patients with hematologic malignancy, including neutropenic patients with sepsis caused by extreme-drug resistant (XDR) strains.

We believe the early switch to ceftolozane/tazobactam at the first clinical suspicion of MDR and early source control were critical factors in the survival of our patients. Among the patient and environmental isolates available for analysis, sequencing confirmed that the *P. aeruginosa* isolates recovered from patients and environmental sources belonged to high-risk ST175 and ST235 clones, which have been linked to the global emergence of MDR strains resistant to fluoroquinolones and a capacity for rapid emergence of resistance to β-lactams and carbapenems through mutation and acquisition of resistance elements [8,9]. These high-risk clones are associated with increased risk of treatment failure due not only to antibiotic resistance, but also their frequent expression of a number of established virulence factors such as type III secretion systems that elaborate potent exotoxins such as the phospholipase ExoU, which is often associated with extensive tissue destruction and localized immunosuppression [11].

In conclusion, the clinical presentation of a high-virulent *Pseudomonas* infection may be an important clinical sign that an isolate is a high-risk clonal subtype predisposed to rapid emergence of MDR, thus supporting early use of ceftolozane/tazobactam in neutropenic patients until susceptibility is established and clinical infection has stabilized.

## Figures and Tables

**Figure 1 microorganisms-08-02055-f001:**
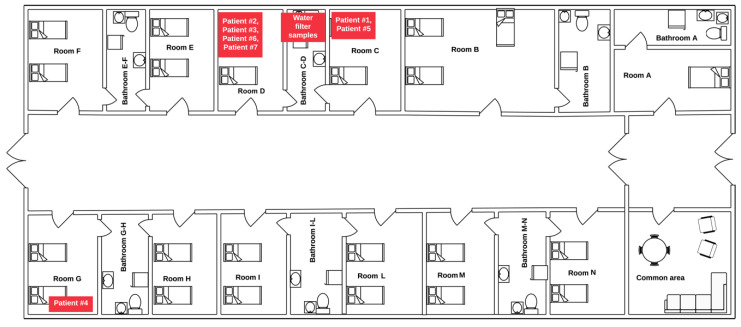
Floorplan of the hematology unit. Six of the reported cases clustered around rooms D and C, which share a bathroom where environmental isolates were collected. One case was detected in a patient from room G.

**Figure 2 microorganisms-08-02055-f002:**
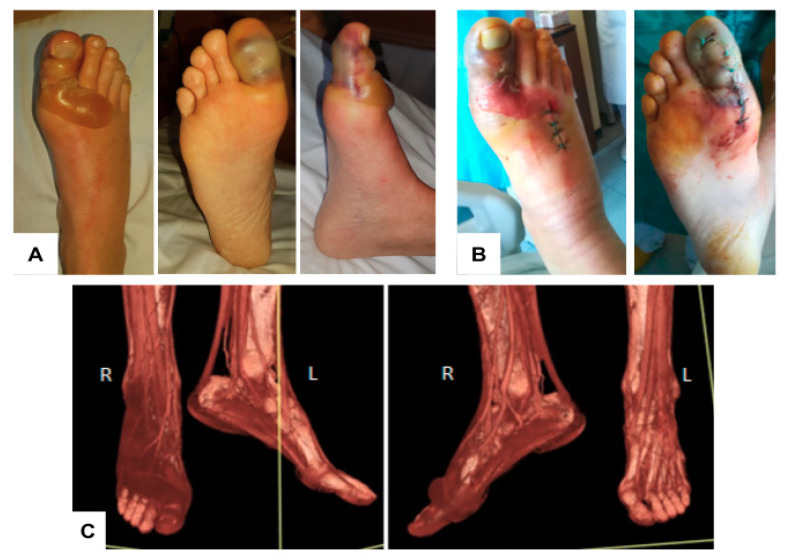
Clinical presentation of *P. aeruginosa* necrotizing fasciitis. (**A**) The small skin lesion rapidly evolved into blisters; (**B**) Appearance of the foot after open drainage attempt; (**C**) Computed tomography (CT) examination of the legs with volume rendering post-processing reformatted images showing the presence of pathologic tissue encasing the vessels, tendons, and muscles of the dorsal region of the right foot and ankle. Only the first toe is involved, while the others are spared.

**Figure 3 microorganisms-08-02055-f003:**
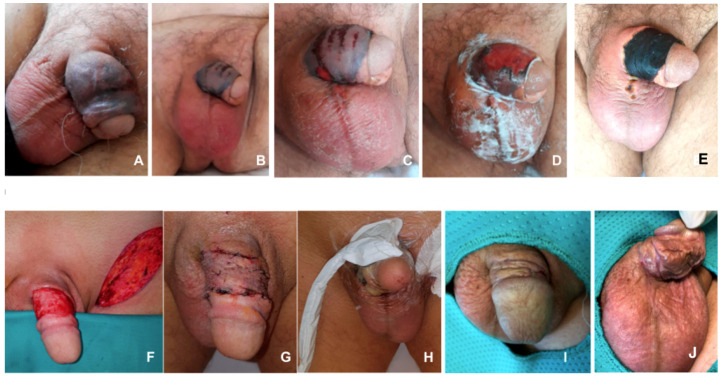
Clinical presentation of *P. aeruginosa* Fournier’s gangrene. (**A**–**D**) Evolving infection of penis and scrotum; (**E**) Development of a dry necrotic dorsal eschar; (**F**) Debridement of necrotic tissue of the penis shaft and full thickness skin taken from left inguinal area; (**G**) Full thickness skin grafted on the soft tissue loss; (**H**) Negative pressure dressing positioned over the skin graft to improve skin graft take (**I**,**J**).

**Figure 4 microorganisms-08-02055-f004:**
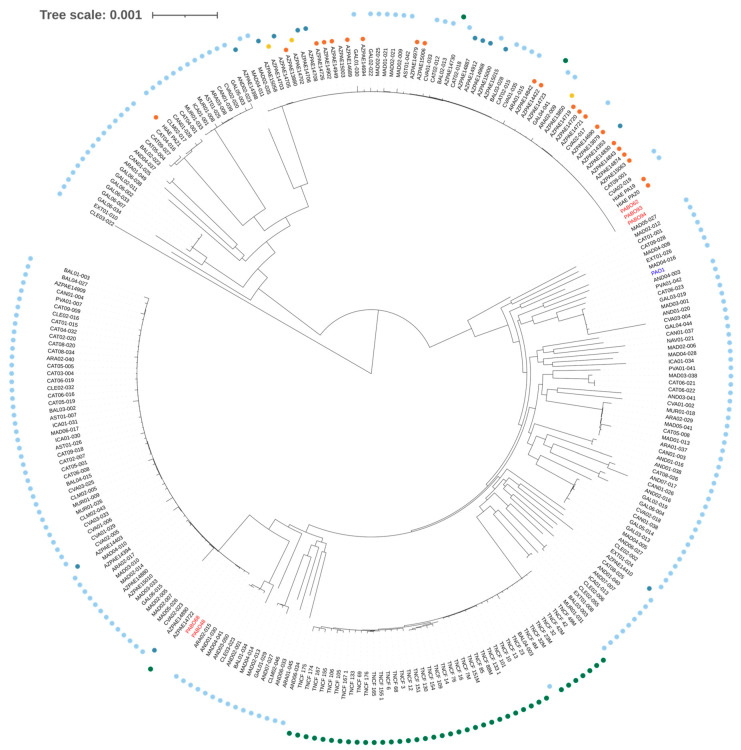
Maximum likelihood phylogeny based on the core genome SNP of the *Pseudomonas aeruginosa* genomes. The *P. aeruginosa* strains (clinical and environmental isolates) included in this study are written in red font, and the references strain is written in blue font. Yellow circles indicate strains isolated in Asia, orange circles indicate strains isolated from North and South America, light blue circles indicate strains isolated from Spain, green circles indicate strains isolated from Italy, and dark blue circles indicate strains isolated from other European Union countries.

**Table 1 microorganisms-08-02055-t001:** Clinical characteristics and outcomes of patients with *P. aeruginosa* infection.

ID	Case 1	Case 2	Case 3	Case 4
Room	C	D	D	G
Isolate (MLST type)	N/A	N/A	PABO62 (ST235)	PABO48 (ST175)
Date of onset	Feb 8	Feb 19	April 7	May 4
Hematological disease	ALL	AML	AML	LY
Age	43	44	60	64
Disease status	CR	Onset	Onset	Relapse
Chemotherapy	Consol.	Induction	Induction	Relapse
Neutropenia	Yes	Yes	Yes	Yes
Antimicrobial prophylaxis	TMP-SMX	TMP-SMX, levofloxacin	TMP-SMX	TMP-SMX, levofloxacin
Site of infection	Blood	Blood, skin, muscle	Blood, genitals	Blood, CVC
Culture site	Peripheral blood	Peripheral blood	Peripheral blood	CVC
Clinical presentation	Severe sepsis	Necrotizing fasciitis	Fournier’s gangrene	Septic shock
Treatment	MER	C/T + COL + AMK + MER; surgery	C/T + COL + AMK + CLI; Surgery	C/T
Outcome	Alive	Alive (leg amputation)	Alive (plastic surgery)	Alive

ALL: acute lymphoblastic leukemia; AML; acute myeloblastic leukemia; CR: complete remission; TMP-SMX: trimethoprim-sulfamethoxazole; AMK: amikacin; CEF: cefepime; PTZ: piperacillin-tazobactam; MER: meropenem; COL: colistin; C/T: ceftolozane-tazobactam; IMP: imipenem; CLI: clindamycin, ASCT: autologous stem-cell transplantation; CVC: central venous catheter; N/A: Not Applicable.

**Table 2 microorganisms-08-02055-t002:** Phenotypic characteristics of *Pseudomonas aeruginosa* strains isolated from patients and the environment.

		MIC (µg/mL)									
Strain (Source)	Sample	CAZ	FEP	IPM	MEM	TZP	CAZ/AVI	C/T	CST	GEN	AMK
B062 (Case #3)	Blood	32	>8	<1	1	>16	4	<1	0.5	>4	16
BO48 (Case #4)	Blood	16	>8	>8	16	>16	4	<1	0.5	>4	16
BO68 (Case #6)	Blood	32	>8	>8	8	>16	4	<1	0.5	>4	<8
BO93 (bidet)	Bidet filter	32	>8	8	16	>16	>8	>16	0.5	>4	>16
BO94 (bidet)	Bidet water	32	>8	>8	16	>16	>8	>16	0.5	>4	>16

Caz, ceftazidime; FEP, cefepime; IPM, imipenem; MEM, meropenem; TZP, piperacillin-tazobactam; CAZ/AVI, ceftazidime/avibactam; C/T, ceftolozane/tazobactam; CST, colistin; GEN, gentamicin; AMK, amikacin.

**Table 3 microorganisms-08-02055-t003:** Genotypic characteristics of *Pseudomonas aeruginosa* strains isolated from patients and the environment.

Strain (Source)	ST	Carbapenemase	Beta-Lactams	Amino-Glycosides	Fluoro-Quinolones	Sulfon-Amide	No. PhageRegions
B062 (Case #3)	235		*bla*OXA-488*, bla*PAO	aac(6′)-Ib3, aadA11, ant(2′′)-Ia, aph(3′)-IIb	*aac(6*′*)-Ib-cr*	*sul1*	9
BO48 (Case #4)	175		*bla*OXA-50, *bla*PAO,	ant(2′′)-Ia, aph(3’)-IIb	*crpP*	*sul1*	11
BO68 (Case #6)	175		*bla*OXA-50, *bla*PAO,	*ant(2*′′*)-Ia, aph(3*′*)-Iib*	*crpP*	*sul1*	11
BO93 (bidet)	235	*bla*VIM-1	*bla*OXA-488, *bla*PAO	*aac(6*′*)-Ib3, aadA1, aadA11, ant(2*′′*)-Ia, aph(3*′*)-IIb*	*aac(6*′*)-Ib-cr*	*sul1*	12
BO94 (bidet)	235	*bla*VIM-1	*bla*OXA-488, *bla*PAO	*Ib3, aadA1, aadA11, ant(2*′′*)-Ia, aph(3*′*)-IIb*	*aac(6*′*)-Ib-cr*	*sul1*	12

ST, sequence type.

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
