# Peer review of "Ceftolozane-Tazobactam Treatment of Hypervirulent Multidrug Resistant Pseudomonas aeruginosa Infections in Neutropenic Patients"

_microorganisms, 2020, doi:10.3390/microorganisms8122055_

Round 1

Reviewer 1 Report

The article is devoted to the urgent problem of treatment of infections caused by multi-resistant and hypervirulent P. aeruginosa. The authors described an outbreak of nosocomial infections caused by such strains in neutropenic patients and demonstrated efficiency of ceftolozane/tazobactam when combined with early source control. However, novelty of the paper is limited. It should also point out significant flaws in the study design and presentation of the material. The description of the results is insufficiently systematized, which significantly complicates the perception.  

  • The authors provide an overly detailed description of the cases, but the microbiological characteristics of the obtained isolates are limited.
    • First, there are no data on MIC’s of antibiotics
    • The authors performed genome-wide sequencing, but only data on virulence genes were provided, data on resistome were not presented, and possible mechanisms of resistance were not analyzed.
    • It is almost impossible to evaluate the results of core genome SNP analysis presented in Figs S1 and S2, since the authors do not report on the origin of genomes used for comparison.
    • Investigation of the pathways of the spread of P. aeruginosa in the department is not done in sufficient detail. According to the data presented, the same type was identified only in one patient and from the bathroom.
    • The authors indicate that isolates from patients 1 and 2 were not available for study. However, the results of the study of isolates from patients 5 and 7 were also not found.
  • Table 2 needs to be revised. The strains differ only in the set of exotoxin genes, sets of all other virulence genes are similar.
  • According to my knowledge Vitek2 is produced by BioMerieux – line 179

The article can be reduced to a short communication format

Author Response

Reviewer #1

The article is devoted to the urgent problem of treatment of infections caused by multi-resistant and hypervirulent P. aeruginosa. The authors described an outbreak of nosocomial infections caused by such strains in neutropenic patients and demonstrated efficiency of ceftolozane/tazobactam when combined with early source control. However, novelty of the paper is limited. It should also point out significant flaws in the study design and presentation of the material. The description of the results is insufficiently systematized, which significantly complicates the perception.  

The authors provide an overly detailed description of the cases, but the microbiological characteristics of the obtained isolates are limited.

Authors’ response: We acknowledge some of the limitations suggested by the reviewer that we have attempted to address as completely as possible in this revision but wish to highlight that clinical recognition of the syndromes presented here, prior to microbiological documentation of resistance and isolate sequencing drove the early empiric switch of therapy to ceftolozane-tazobactam and recognition of probable common source. We believe the clinical aspects of the case are a core part of the message of the case series.

Reviewer comment:

  1. First, there are no data on MIC’s of antibiotics

Authors’ response: We have added a new table containing MIC and genotypic data in the manuscript (see new Table 2).

  1. The authors performed genome-wide sequencing, but only data on virulence genes were provided, data on resistome were not presented, and possible mechanisms of resistance were not analyzed.

Authors’ response: Data regarding antimicrobial resistance genes were added to the text as requested (see new Table 3).

  1. It is almost impossible to evaluate the results of core genome SNP analysis presented in Figs S1 and S2, since the authors do not report on the origin of genomes used for comparison.

Authors’ reply and amendments: To better define the clonal relatedness of the P. aeruginosa strains included in this study and origins of the genomes, we prepared a complete phylogenetic tree based on the genomes obtained from isolate sequences registered in Genbank and banked P. aeruginosa in our clinical microbiology laboratory-see new Figure 4.  

  1. Investigation of the pathways of the spread of aeruginosa in the department is not done in sufficient detail. According to the data presented, the same type was identified only in one patient and from the bathroom.

Authors’ reply and amendments: We have provided more details on the investigations performed to identify the common source in lines 176-181. Our analysis was limited by the fact that isolates from the first two cases were not available for analysis by the time an outbreak was suspected. While it is true that only one isolate with the same type from the water trap was detected in a patient (case#3); 6/7 remaining patient cases of MDR P. aeruginosa described in this report were clustered around a single shared bathroom in the hematology unit, which is highly suggestive of a common source for the infections.

  1. The authors indicate that isolates from patients 1 and 2 were not available for study. However, the results of the study of isolates from patients 5 and 7 were also not found.

Authors’ reply: Cases 5-6 manifested with less-severe infection so they were not initially suspected to be part of common source outbreak of the hypervirulent strain and isolated were not stored. Patient #7 developed the P. aeruginosa sepsis at an outside hospital two months after discharge from room D.

  1. Table 2 needs to be revised. The strains differ only in the set of exotoxin genes, sets of all other virulence genes are similar.

Authors’ reply: We agree the table is a little repetitive, so we have removed table 2 and now summarize the different virulence gene data in the text in lines 247-259.

  1. According to my knowledge Vitek2 is produced by BioMerieux – line 179

Authors’ reply: We apologize for the mistake. We modified the instrument used by correcting the instruments name (see line 190-198)

  1. The article can be reduced to a short communication format

Authors’ reply: Given that the reviewer has recommended more detailed microbiological information (now added) and our assertion that the clinical presentation is an essential component of the manuscript, we believe shortening the manuscript to a short communication is unlikely to improve manuscript clarity and interest to readers.

Reviewer 2 Report

This paper describes an outbreak of MDR P. aeruginosa and the successful treatment of 7 patients. It is well written and constructed.

Please include the MICs to Ceftolozane-tazobactam.

It would be useful to know what percentage of invasive Pseudomonas isolates in your hospital setting are susceptible to Ceftol/Tazo. While you report 90% susceptibility in other studies, how useful is the drug in your setting?

Please mention side-effects of the drug or lack thereof.

Check spelling (form - from), italics for P. aeruginosa and sentence structure.

Author Response

This paper describes an outbreak of MDR P. aeruginosa and the successful treatment of 7 patients. It is well written and constructed.

  1. Please include the MICs to Ceftolozane-tazobactam.

Authors’ reply: The phenotypic and genotypic data of P. aeruginosa strains included in this study were added to text, as requested (see new Tables 2 & 3)

  1. It would be useful to know what percentage of invasive Pseudomonas isolates in your hospital setting are susceptible to Ceftol/Tazo. While you report 90% susceptibility in other studies, how useful is the drug in your setting?

Authors’ reply: We have added a statement to this effect at the beginning of the methods/microbiology section, lines 193-194: “Previous cumulative susceptibility reports from our institution indicated 92% of all tested P. aeruginosa isolates (96% from blood samples) were susceptible to ceftolozane/tazobactam”

  1. Please mention side-effects of the drug or lack thereof.

Authors’ reply: We have added a brief statement in the introduction, lines 53-54 indicating the safety profile is similar to other approved cephalosporins

  1. Check spelling (form - from), italics for P. aeruginosa and sentence structure.
  2. Authors’ reply: We carefully reviewed and spellchecked the final document.

Round 2

Reviewer 1 Report

Authors significantelly improved manuscropt. I have no significant remarks and comments.